# A Comparison between Open and Minimally Invasive Techniques for the Resection of Colorectal Liver Metastasis

**DOI:** 10.3390/healthcare10122433

**Published:** 2022-12-02

**Authors:** Ilenia Bartolini, Laura Fortuna, Matteo Risaliti, Luca Tirloni, Simone Buccianti, Cristina Luceri, Maria Novella Ringressi, Giacomo Batignani, Antonio Taddei

**Affiliations:** 1Department of Experimental and Clinical Medicine, AOU Careggi, 50134 Florence, Italy; 2Department of Neuroscience, Psychology, Drug Research and Child Health—NEUROFARBA, University of Florence, 50134 Florence, Italy

**Keywords:** liver metastasis, colorectal cancer, minimally invasive surgery, implementation of liver surgery

## Abstract

The liver is the most common site of colorectal cancer metastasis. Liver surgery is a cornerstone in treatment, with progressive expansion of minimally invasive surgery (MIS). This study aims to compare short- and long-term outcomes of open surgery and MIS for the treatment of colorectal adenocarcinoma liver metastasis during the first three years of increasing caseload and implementation of MIS use in liver surgery. All patients treated between November 2018 and August 2021 at Careggi Teaching Hospital in Florence, Italy, were prospectively entered into a database and retrospectively reviewed. Fifty-one patients were resected (41 open, 10 MIS). Considering that patients with a significantly higher number of lesions underwent open surgery and operative results were similar, postoperative morbidity rate and length of hospital stay were significantly higher in the open group. No differences were found in the pathological specimen. The postoperative mortality rate was 2%. Mean overall survival and disease-free survival were 46 months (95% CI 42–50) and 22 months (95% CI 15.6–29), respectively. The use of minimally invasive techniques in liver surgery is safe and feasible if surgeons have adequate expertise. MIS and parenchymal sparing resections should be preferred whenever technically feasible.

## 1. Introduction

Colorectal cancer (CRC) is one of the most common malignant neoplasms in the world and the third leading cause of cancer-related deaths in developed countries [1]. In addition to patients presenting with distant metastasis (about 20–34%) [2], it is estimated that 20–50% of patients who undergo curative colorectal resection and eventually perioperative therapies will develop a metachronous recurrence [3,4]. The liver is the most common site for the distant spread of CRC. Liver metastases occur in up to 60% in an asynchronous (13–25%) or a metachronous (up to 75%) manner [5,6].

Within multimodality and multidisciplinary treatment, surgery is a cornerstone in the management of CRC metastases whenever technically feasible and oncologically appropriate [7]. Due to technical and technological improvements and perioperative care, indications for liver resection have been widened over the past three decades, maintaining acceptable morbidity and mortality rates [8]. Unfortunately, only about 25% of the metastatic patients can be resected with negative margins, preserving an adequate liver remnant volume [6].Despite a high recurrence rate—up to 80%—with a 10–15% chance of early recurrence and disease-specific deaths for these patients, the expected 5-year survival rate reaches about 40–74% and the cure rate reaches about 20% [8,9].

Concerns surrounding laparoscopic liver resection have been gradually overcome owing to the emerging benefits offered by MIS in short-term outcomes [10,11] with at least similar oncologic outcomes (R0 resections, tumor recurrence, OS, and DFS) in high-volume centers [11,12,13,14,15]. Several papers, including reviews, meta-analyses, and the randomized controlled trial OSLO-COMET [16], suggested reduced intraoperative blood loss, lower morbidity rates, and a shorter length of hospital stay for MIS compared to open surgery for colorectal liver metastasis [13,14,15,17].

Furthermore, after the first laparoscopic hepatectomies reported in the 1990s [18], from the Louisville statement in 2008 to the Southampton statement in 2017, indications for MIS have been greatly widened and changed [12,19,20]. Initially, MIS was only indicated to treat solitary lesions smaller than 5 cm and located in segments 2 to 6 (the anterolateral segments or the so-called laparoscopic segments) [19]. The Morioka consensus conference in 2014 focused on the comparison between MIS and open surgery, trying to demonstrate the non-inferiority of MIS [20]. The last statement moved toward emphasizing the benefits of MIS over open surgery and tried to promote MIS-safe implementation [12]. Older age, high BMI, previous liver resections, combined resection of the primary and metastatic disease, or complex resections near pedicles or in the posterosuperior segments (1, 4a, 7, and 8) are no longer contraindications to MIS in experienced hands [12,21].

In 2003, Giulianotti et al. [22] reported their first experience in robotic liver resections. Although a faster expansion was demonstrated, along with good safety and feasibility profile [23], a real superiority of the robotic approach over laparoscopy has not yet been widely found [12,24,25]. The higher costs and longer operative time are again the most frequently cited drawbacks. Nevertheless, the robotic platform could provide a lower rate of R1 and wider margins, especially in difficult procedures [26], and may shorten the learning curve, allowing good results even in lower-volume centers [23].

Even if up to 70% of the patients with colorectal liver metastasis could be a candidate for MIS in high-volume centers [27], minimally invasive techniques, though frequently used in many operations on the digestive tract, are still far from being considered a gold standard in liver surgery, being used in about 20% of patients with liver cancer [28]. Furthermore, in liver surgery there is quite a steep learning curve and at least 20 MIS procedures are required, though this number has been decreasing over recent years [29]. This study aims to compare open surgery and MIS in the treatment of the first occurrence of colorectal adenocarcinoma liver metastasis in terms of short- and long-term outcomes during the first three years of a progressive case-volume increase. It will examine the implementation of MIS in liver surgery to demonstrate its safety and feasibility.

## 2. Materials and Methods

In 2018, a program of progressive organization of different fields of general surgery between the different surgical units began at Careggi Teaching Hospital in Florence, Italy. All patients undergoing surgery between November 2018 and August 2021 for a first occurrence of liver metastasis from colorectal adenocarcinoma were prospectively entered into a dedicated database containing patient information, treatment data, results of the pathological examination, and long-term oncological outcomes.

Patient selection is represented in Figure 1.

To compare the techniques within a homogenous group of patients, primary tumors different from colorectal adenocarcinoma or patients already treated for liver metastasis before the study period were excluded from the analysis.

Preoperative data comprehended demographic information and medical history including the body mass index (BMI), previous abdominal surgery or preoperative chemo or radiotherapy, and the results of the preoperative evaluations. Routine preoperative assessment included triple-phase contrast-enhanced Computed Tomography and MRI with an organ-specific contrast medium. A liver biopsy of the lesion was required in doubtful cases or when a histological diagnosis was required to start chemotherapy. In selected cases, a Positron Emission Tomography scan was requested. Carcinoembryonic antigen (CEA) and carbohydrate antigen 19.9 (CA 19.9) were also evaluated.

Liver metastasis presentation was considered metachronous if it occurred at least 3 months after the diagnosis of the primary tumor.

Treatment indications for each patient were given following the Multidisciplinary Team evaluation. The surgery type and technique were chosen by surgeons with expertise in both liver surgery and minimally invasive techniques. Each resection was performed by experienced surgeons or under their direct supervision.

Major hepatectomies were defined as the resection of at least three segments according to Brisbane’s classification [30].

All patients received prophylactic antibiotic therapy and low-weight molecular heparin to prevent site infections and deep-venous thromboembolism, respectively. The preoperative planning was confirmed after the exploration of the abdominal cavity and after intraoperative ultrasonography evaluation.

The operative room set-up is represented in Figure 2.

Three surgeons (two attendings and one resident) were usually involved in both open surgery and MIS. Parenchymal transection was performed with the Cavitron Ultrasonic Aspirator (CUSA) for the open and laparoscopic technique, or preferably with monopolar scissor and bipolar grasp for the robotic technique. In MIS, the transparenchymal approach was the most used, especially in non-anatomical resections. A Pfannenstiel incision was most often utilized to extract the specimen after MIS.

The international normalized ratio (INR) and bilirubin level were evaluated on postoperative day 5 according to the “50–50” criterion [31]. Clavien–Dindo’s model was used to classify postoperative complications [32]. Mortality was defined as a 90-day or in-hospital surgery-related death. The disease-free interval was considered to be the time between liver surgery or colorectal surgery in the case of liver-first treatment and the diagnosis of any site of recurrence of disease or until the date of death. Overall survival was considered to be the time between the colorectal surgery or liver surgery in the case of liver-first treatment and the date of death or the last visit for living patients. Recurrences were treated with surgery, chemotherapy, radiotherapy, percutaneous treatment, combinations of the aforementioned, or best supportive care as appropriate.

Follow-ups were conducted in a multidisciplinary manner involving surgeons and oncologists. Data were collected from medical files or updated by phone call.

All data were prospectively collected and retrospectively reviewed.

The analysis was conducted with an “intention-to-treat” aim.

Quantitative data are expressed as median and interquartile range values. Comparisons based on quantitative data were performed using the Mann–Whitney test while categorical variables were compared using the χ2 or Fisher’s exact test. The inverse probability of treatment weights was calculated using a logistic regression model including variables selected as potential confounders for the relationship between surgical approach and clinical outcomes (age, BMI, smoking, comorbidity, previous abdominal surgery, chemotherapy (CHT) before surgery, and the number of lesions). Missing data were explicitly mentioned in the tables if they resulted in more than 10% of the total. Statistical significance was defined as a *p*-value < 0.05.

An estimate of DFS and OS rates was calculated according to the Kaplan–Meier method and compared using the Log-rank test.

All collected information was analyzed using the SPSS for Windows 24.0 software package (SPSS Inc, Chicago, IL, USA).

This study was approved by the Institutional Ethics Committee of the Area Vasta Centro (protocol number 22397).

## 3. Results

During the study period, 51 patients underwent surgery for the first occurrence of colorectal adenocarcinoma liver metastasis. One patient had a stroke on postoperative day 3, causing death, and was therefore excluded from the short-term outcomes analysis. A total of 40 patients were treated with an open technique and 10 with a minimally invasive approach (6 laparoscopically assisted and 4 robotic assisted).

Patient characteristics are shown in Table 1.

Previous abdominal surgery does not represent a contraindication to MIS as the great majority of the patients treated with MIS had a previous abdominal intervention in their medical history.

No significant differences were found in patient characteristics. However, a greater number of patients receiving medical treatment before liver surgery were treated with the open technique. Before liver surgery, the majority of the patients were treated, eventually within clinical trials, with the triplet FOLFOX (folinic acid, fluorouracil, and oxaliplatin) or with the quadruplet FOLFOXIRI (folinic acid, fluorouracil, oxaliplatin, and irinotecan). Furthermore, most of them received a biological agent (i.e., cetuximab, bevacizumab, or panitumumab). A restaging imaging (CT scan and/or MRI) was always performed. A stable disease was the minimal requirement to proceed to surgery, except for selected patients experiencing disease progression (but who were still resectable) and therefore deemed unfit for further lines of medical treatment.

Tumor characteristics are shown in Table 2.

One patient was treated for lung and liver metastasis secondary to colorectal cancer, but the site of the primary tumor is still unknown. Two patients did not undergo surgery for their primary tumors because of disease progression.

There were no significant differences in the TNM stage of the primary tumor or in the time of metastasis presentation between the two groups. The number of liver lesions was significantly higher in the open group, and there was a higher percentage of lesions located in the so-called laparoscopic segments [19] in the MIS group (Figure 3).

In the MIS group, there were no lesions in segment 1 and there was a higher prevalence of metastasis located in segment 5. However, it is not only the mere segment location of a lesion that leads the surgeon to choose what kind of technique to use but also the number of lesions and their relation to the liver pedicles and veins.

The operative results are shown in Table 3.

The conversion to open surgery rate was 10%, and only one patient required conversion from the laparoscopic technique due to of a high grade of liver steatosis. No conversion from robotic to laparoscopic technique was performed.

Although not significant, patients needing a major resection were more frequently treated with the open technique. Surgery time was similar for both open and MIS techniques, but the difference in surgery complexity must be considered. The patient undergoing the longest operation had both the primary tumor and the metastasis resected.

An attempt to prepare the hepatic hilum for the Pringle maneuver was always performed in both open surgery and MIS. However, the Pringle maneuver was used more frequently with the open technique. The hanging maneuver and vascular resections were less frequently, but not significantly, performed in the MIS group. Intraoperative complications included three respiratory, three vascular, and one biliary complication.

Table 4 shows the postoperative results.

Postoperative complications were observed only in the open group. Mild complications included perihepatic fluid collections, mild and transient postoperative liver failure, chylous ascites, or surgical site infection; severe complications included biliary leak or pleural effusion causing respiratory distress needing a percutaneous drain (some of them appeared after discharge, thus requiring readmission). Only one patient experiencing a high-flow biliary leak required a redo-surgery (right hepatectomy) after multiple wedge resections. Hospital stay was significantly lower in the MIS group.

Pathological results and oncological outcomes are shown in Table 5.

No significant differences were found in the results of the pathological specimen analysis. The global positive margin rate was 4%. No R1 vascular resections were reported.

Mutations in BRAF, NRAS, or in other genes (for example, PIK3CA), and microsatellite stability/instability were not frequently detected. The regimens used for adjuvant chemotherapy were the same as those used before liver surgery if no disease progression had occurred. The time to start CHT was not significantly different between the two groups.

The most frequent sites of recurrence were the liver followed by the lung, perianastomotic site, and peritoneum. Five of the thirty patients experiencing recurrence (all treated with an open liver surgery) were disease-free after further treatments (chemotherapy and surgical resection).

After analyzing the initial group of 51 patients, postoperative mortality was 2%. With a median follow-up of 25 months (1–52 months), the estimated mean OS was 46 months (95% CI 42–50). All deceased patients were treated with an open technique, thus precluding further comparative analysis. The estimated mean DFS was 22 months (95% CI 15.6–29).

Figure 4 shows the Kaplan–Meier curve of DFS stratified by technique.

In performing a further analysis stratifying for the number of lesions (1–2 vs. 3 or more), no significant differences were found in the DFS (Figure 5). Similarly, no differences were found stratifying the analysis for the technique used to treat primary cancer (*p* = 0.148), margin status (*p* = 0.153), and KRAS mutation (*p* = 0.735) (Figure 6).

Finally, the inverse probability of treatment weights was calculated including variables selected as potential confounders for the relationship between surgical approach and clinical outcomes. The odds ratios (and 95% CI) for recurrence, the persistence of disease, and survival were analyzed using generalized estimating equations, incorporating the weights and using open surgery as the reference group. MIS reduced the risk of recurrence by about nine times compared to open surgery (OR = 0.107, CI 0.014–0.841; *p* = 0.034) but did not modify other clinical outcomes.

## 4. Discussion

The present study reports our recent surgical experience with a selected cohort of patients treated during the first three years of a progressive case-volume increase and implementation of MIS in liver surgery by surgeons with previous expertise in both liver surgery and the use of minimally invasive techniques. Consequently, these results could reflect the last part of the proficiency step of the learning curve. The attempt to standardize the technique in MIS is of paramount importance to achieve an adequate level of care, most of all in technically demanding resections (e.g., the resection of segment 8) [33]. This is a continuously evolving process that involves all steps of the surgical intervention including patient selection, OR setup, each procedure performed during surgery, and the anesthesiologic aspect. Some of these aspects have been reported in the Materials and Methods section while further technical details are related to each different procedure and are of no object to this paper.

In this series, minor and non-anatomical resections were performed in most of the patients with higher (but not significant) percentages in the MIS group. Unlike hepatocellular hepatocarcinoma, for colorectal metastasis, parenchymal sparing surgery seems to be preferable to anatomic resection whenever possible [12,34,35,36]. As colorectal metastasis could be considered a kind of chronic disease, preserving more parenchyma may allow further resections [14].

Significantly greater use of the Pringle maneuver was found during open surgery. Intermittent Pringle maneuvers or continuous hemi-hepatic inflow control do not impair liver function and can be used if necessary, but it is not mandatory [12,37].

Only patients treated with the open technique experienced postoperative complications and the global morbidity rate was 42.5%. However, the higher burden of disease requiring more extended resections in the open group could explain this result. Consistently, similar to the previously cited papers, the length of hospital stay was significantly lower in the MIS group in our series.

Reported postoperative morbidity rates were about 23% and 44% for MIS and open surgery, respectively [14]. Interestingly, an inverse correlation between morbidity and survival has been proposed. Possible explanations include a prolonged phase of immunosuppression and a delayed start of chemotherapy [38,39].

The surgical status margin is an important prognostic factor for disease recurrence [40,41]. A margin width of 1 mm seems sufficient to ensure good DFS rates while wider margins do not confer a greater survival benefit [40]. Our pathological results reported a global R1 resection rate of 4% (2.5% and 10% for the open and MIS groups, respectively). A possible explanation for this could involve the different parenchymal transection methods. Conversely, reported positive margin rates were 25% and 7% for open and MIS surgery, respectively [14].

Most of our patients received a perioperative chemotherapy treatment, generally platinum-based, and most of them were also treated with biological drugs. Perioperative chemotherapy may allow for a prolonged DFS in both the setting of resectable or upfront unresectable disease [42]. However, potential hepatotoxicity derived from chemotherapy (mostly platinum-related sinusoidal obstruction syndrome) must be considered when evaluating liver resection extension to reduce the risk of post-hepatectomy liver failure. On the contrary, the inhibitor of the vascular endothelial growth factor bevacizumab seems to protect the liver from this damage [43].

This study has some limitations. It is a non-randomized, retrospective study with an inherent selection bias. The series is quite small, and this should lead to a careful and critical interpretation of some findings. Nevertheless, the small number of patients analyzed precluded a propensity score-matched analysis that could add value to the results. Some missing data may also cause bias throughout the analysis. Although it is a peculiarity of the paper, the fact that outcomes reflect the proficiency step of the learning curve cannot allow for a generalization of the results. There are different instruments used with laparoscopic and robotic techniques that are likely to introduce confounding factors in the analysis. Although part of a recent series, the follow-up period is quite short and consequently further longer-term analyses are required.

In conclusion, although the evidence from this study is too weak to draw definitive judgments, the use of minimally invasive techniques in liver surgery is safe and feasible if the surgeons have adequate expertise. Nonetheless, the skillfulness of the surgeons and the multidisciplinary evaluation is of paramount importance to providing patients with the best treatment. MIS and parenchymal sparing resections should be preferred whenever technically feasible, providing better short-term outcomes and similar oncologic results compared to open surgery. Since a randomized controlled trial could be difficult at an ethical level, larger numbers are required to perform at least a propensity-score matched analysis.

## Figures and Tables

**Figure 1 healthcare-10-02433-f001:**
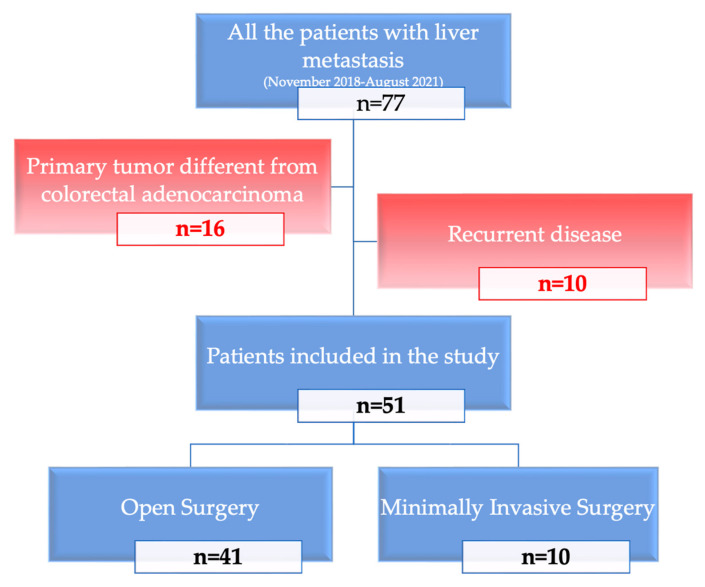
Flowchart representing the process of patient selection. In red: exclusion criteria.

**Figure 2 healthcare-10-02433-f002:**
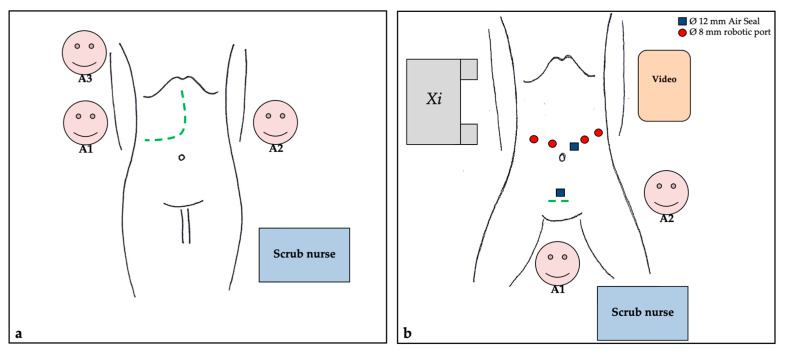
Operative room set-up. (**a**). Open technique. (**b**). Robotic/laparoscopic technique. The hypogastric trocar is used for the Pringle maneuver.

**Figure 3 healthcare-10-02433-f003:**
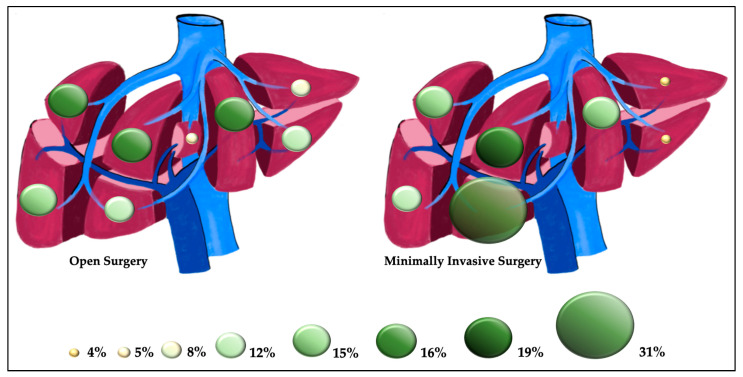
Proportional distribution of the metastasis for the two groups.

**Figure 4 healthcare-10-02433-f004:**
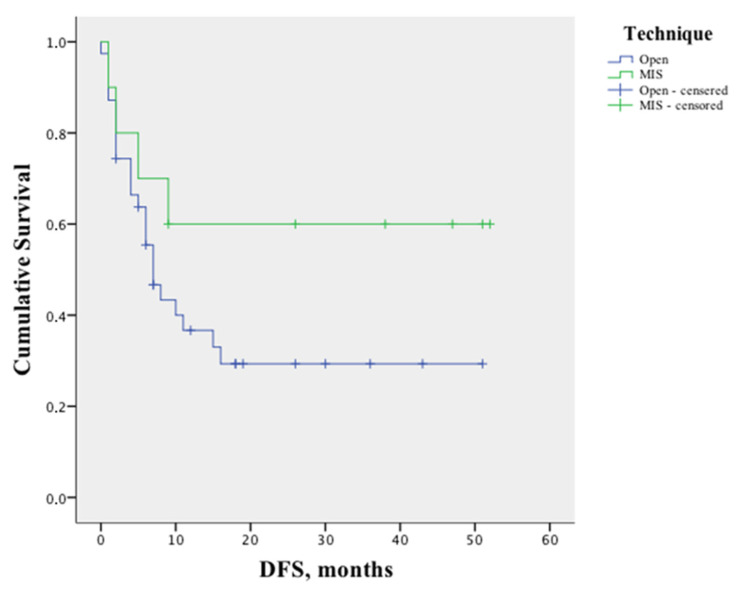
Kaplan–Meier curve of disease-free survival (DFS) (*p* = 0.164) stratified by technique of liver resection.

**Figure 5 healthcare-10-02433-f005:**
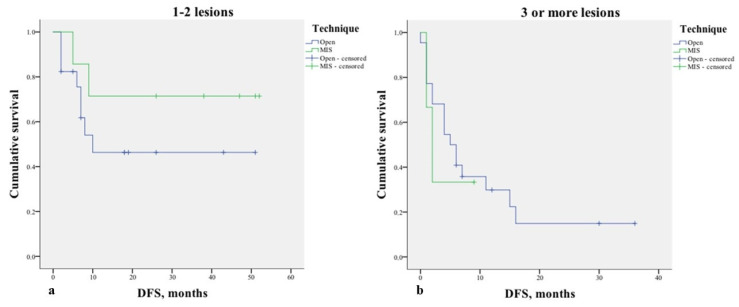
Kaplan–Meier curve of disease-free survival (DFS) (*p* = 0.527) stratified by the preoperative number of resected lesions: (**a**). 1–2 lesions. Mean DFS rates for open surgery and MIS were 26.7 months ±6 and 39.1 months ±7.7, respectively; (**b**). 3 or more lesions. Mean DFS rates for open surgery and MIS were 10.4 months ±2.7 and 4 months ±2, respectively.

**Figure 6 healthcare-10-02433-f006:**
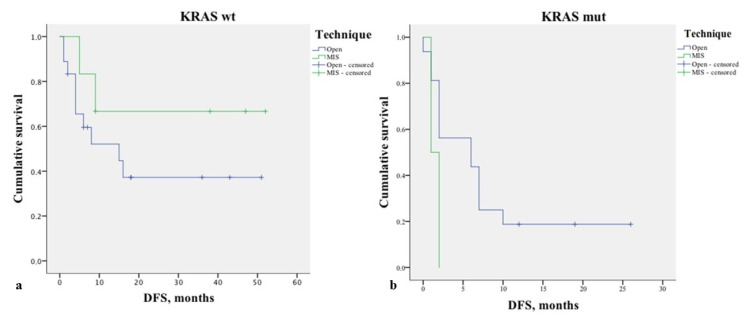
Kaplan–Meier curve of disease-free survival (DFS) (*p* = 0.735) stratified by KRAS status. (**a**). KRAS wild type. Mean DFS rates for open surgery and MIS were 23.2 months ±5.6 and 37 months ±8.7, respectively; (**b**). KRAS mutation. Mean DFS rates for open surgery and MIS were 8.2 months ±2.2 and 1.5 months ±0.5, respectively.

**Table 1 healthcare-10-02433-t001:** Patient characteristics.

	Open n = 40 (80%)	MIS n = 10 (20%)	Total n = 50	*p* Value
Age (years)	63 (54.5–71.5)	65 (53–69)	63.5 (54–70)	0.839
BMI	25 (22.5–28)	22.5 (21.5–28)	25 (22–28)	0.978
Gender (n, %)				0.382
Male	26 (65%)	5 (50%)	31 (62%)	
Female	14 (35%)	5 (50%)	19 (38%)	
Smoking habit (n, %)				0.721
No	26 (66.7%)	6 (60%)	32 (65.3%)	
Yes	13 (33.3%)	4 (40%)	17 (34.7%)	
Comorbidities (n, %)				0.567
No	16 (40%)	5 (50%)	21 (42%)	
Yes	24 (60%)	5 (50%)	29 (58%)	
Charlson Comorbidity Index	8 (7–9)	8 (7–9)	8 (7–9)	0.730
ASA (n, %)				0.307
1	1 (2.5%)	1 (10%)	2 (4%)	
2	29 (72.5%)	5 (50%)	34 (68%)	
3	10 (25%)	4 (40%)	14 (28%)	
Previous abdominal surgery (n, %)				0.563
No	7 (17.5%)	1 (10%)	8 (16%)	
Yes	33 (82.5%)	9 (90%)	42 (84%)	
Preoperative Hemoglobin (g/dL)	12.7 (11.8–14.3)	14.4 (13.7–14.7)	13.5 (12–14.5)	0.78
Preoperative Creatinine (mg/dL)	0.79 (0.68–0.90)	0.87 (0.67–0.90)	0.79 (0.68–0.90)	0.839
Preoperative bilirubin (mg/dL)	0.4 (0.3–0.6)	0.4 (0.4–1.1)	0.4 (0.3–0.6)	0.184
Maximum preoperative CEA (ng/mL)	13.7 (5.2–56)	22.8 (3-NE)	17 (5–57)	0.120
RT/CHT before liver surgery (n, %)				0.083
No	9 (22.5%)	5 (50%)	14 (28%)	
Yes	31 (77.5%)	5 (50%)	36 (72%)	
Biological drugs	14 (45%)	3 (60%)	17 (47%)	0.828

MIS = minimally invasive surgery; ASA = American Society of Anesthesiologists; CEA = carcinoembryonic antigen; RT/CHT = include radiotherapy (associated or not with capecitabine) and/or chemotherapy; NE = not evaluable.

**Table 2 healthcare-10-02433-t002:** Tumor characteristics.

	Open n = 40 (80%)	MIS n = 10 (20%)	Total n = 50	*p* Value
Site of the primary tumor (n, %)				0.201
Right colon	8 (20.5%)	3 (30%)	11 (22.4%)	
Left colon	20 (51.3%)	2 (20%)	22 (44.9%)	
Rectum	11 (28.2%)	5 (50%)	16 (32.7%)	
Technique for primary tumor (n, %)				0.336
Minimally invasive	21 (56.8%)	8 (80%)	29 (61.7%)	
MIS Converted to open	4 (10.8%)	0	4 (8.5%)	
Open	12 (32.4%)	2 (20%)	14 (29.8%)	
T stage of the primary tumor				0.345
T 2	3 (8%)	2 (20%)	5 (10.5%)	
T3	30 (81%)	8 (80%)	38 (81%)	
T4	4 (11%)	0	4 (8.5%)	
N stage of the primary tumor				0.493
N0	15 (40.5%)	6 (60%)	21 (45%)	
N1	14 (38%)	2 (20%)	16 (34%)	
N2	8 (21.5%)	2 (20%)	10 (21%)	
Metastasis presentation (n, %)				0.528
Metachronous	15 (37.5%)	6 (60%)	21 (42%)	
Synchronous, bowel-first	12 (30%)	1 (10%)	13 (26%)	
Synchronous, liver-first	9 (22.5%)	2 (20%)	11 (22%)	
Synchronous, combined	4 (10%)	1 (10%)	5 (10%)	
Number of lesions	3 (1–7)	1 (1–3)	2.5 (1–5)	0.046
Maximum diameter (mm)	30 (14–49)	26.5 (22–43.5)	29 (15–45)	0.654
“Laparoscopic segments” only (n, %)				0.197
No	33 (82.5%)	6 (60%)	39 (78%)	
Yes	7 (17.5%)	4 (40%)	11 (22%)	

MIS = minimally invasive surgery.

**Table 3 healthcare-10-02433-t003:** Operative results.

	Open n = 40 (80%)	MIS n = 10 (20%)	Total n = 50	*p* Value
Type of surgery (n, %)				0.178
Major	18 (45%)	2 (20%)	20 (40%)	
Minor	15 (37.5%)	7 (70%)	22 (44%)	
Multiple wedges	7 (17.5%)	1 (10%)	8 (16%)	
Pringle maneuver (n, %)				0.014
No	12 (30%)	8 (80%)	20 (40%)	
Yes	28 (70%)	2 (20%)	30 (60%)	
Number of maneuvers	1 (0–3)	0 (0–1.5)	0.5 (0–3)	0.769
Maximum length (min)	10 (0–15)	0 (0–4.5)	8 (0–15)	0.116
Total length (min)	18.5 (0–34)	0 (0–54.5)	12.5 (0–35)	0.525
Hanging maneuver (n, %)				0.279
No	22 (55%)	8 (80%)	30 (60%)	
Yes	18 (45%)	2 (20%)	20 (40%)	
Vascular resection (n, %)				0.372
No	37 (92.5%)	10 (100%)	47 (94%)	
Yes	3 (7.5%)	0	3 (6%)	
Intraoperative complications (n, %)				0.616
No	35 (87.5%)	8 (80%)	43 (86%)	
Yes	5 (12.5%)	2 (20%)	7 (14%)	
Surgery time (min)	293.5 (240.5–345)	317.5 (183–409)	297.5 (234–360)	0.458

MIS = minimally invasive surgery; Minor = minor hepatectomies/hepatic wedge resections (up to 2); Major = major hepatectomies; Multiple wedges = at least 3 wedges.

**Table 4 healthcare-10-02433-t004:** Postoperative results.

	Open n = 40 (80%)	MIS n = 10 (20%)	Total n = 50	*p* Value
ICU				0.331
No	5 (12.5%)	3 (30%)	8 (16%)	
Yes	35 (87.5%)	7 (70%)	42 (84%)	
INR POD 5	1.1 (1.1–1.2)	1.1 (1–1.1)	1.1 (1.1–1.2)	0.486
Total Bilirubin POD 5 (mg/dL)	0.9 (0.5–1.3)	0.6 (0.5–1.4)	0.9 (0.5–1.4)	0.589
Refeeding (POD)	2 (1–3)	1.5 (1–4)	2 (1–3)	0.966
Bowel function (POD)	4 (3–5)	4 (3.5–5)	4 (4–5)	0.700
Drain removal (POD)	4 (4–7)	3.5 (2–5)	4 (3–6)	0.099
Complications				0.010
No	23 (57.5%)	10 (100%)	33 (66%)	
Yes	17 (42.5%)	0	17 (34%)	
Clavien–Dindo’s I-II	7 (41%)	0	7 (41%)	
Clavien–Dindo’s III-IV	10 (59%)	0	10 (59%)	
Length of hospital stay (days)	7 (5–11)	4.5 (2–7)	6.5 (5–10)	0.012
Need for blood transfusion (n, %)				0.416
No	29 (72.5%)	9 (90%)	38 (76%)	
Yes	11 (27.5%)	1 (10%)	12 (24%)	
Readmission				0.327
No	34 (85%)	10 (100%)	44 (88%)	
Yes	6 (15%)	0	6 (12%)	

MIS = minimally invasive surgery; ICU = intensive care unit; INR = international normalized ratio; POD = postoperative day; CD III - IV = Clavien–Dindo’s Classification grades III–IV; CHT = chemotherapy.

**Table 5 healthcare-10-02433-t005:** Pathological results and other oncological outcomes.

	Open n = 40 (80%)	MIS n = 10 (20%)	Total n = 50	*p* Value
Margin (n, %)				0.363
No	39 (97.5%)	9 (90%)	48 (96%)	
Yes	1 (2.5%)	1 (10%)	2 (4%)	
KRAS mutation				0.269
No	19 (51%)	6 (75%)	25 (56%)	
Yes	18 (49%)	2 (25%)	20 (44%)	
Missing	3	2	5	
Other molecular mutation				0.168
No	27 (73%)	8 (100%)	35 (78%)	
Yes	10 (27%)	0	10 (22%)	
Missing	3	2	5	
CHT after liver surgery (n, %)				0.720
No	16 (41%)	3 (30%)	19 (38.8%)	
Yes	23 (59%)	7 (70%)	30 (61.2%)	
Biological drugs	8 (35%)	1 (14%)	9 (30%)	0.300
Time to start CHT (days)	63 (42.5–78.5)	61.5 (49–75)	63 (46–74)	0.668
Recurrence (n, %)				0.171
No	14 (35%)	6 (60%)	20 (40%)	
Yes	26 (65%)	4 (40%)	30 (60%)	

MIS = minimally invasive surgery; other molecular mutations included a mutation in BRAF, NRAS, or other genes (for example, PIK3CA), and microsatellite stability/instability.

## Data Availability

Dataset can be requested from the Corresponding Author.

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
