# Peer review of "A Comparison between Open and Minimally Invasive Techniques for the Resection of Colorectal Liver Metastasis"

_healthcare, 2022, doi:10.3390/healthcare10122433_

Round 1

Reviewer 1 Report

I have attached a file with my review

Author Response

Thanks for the comments.

  1. The number of patients. ... May this problem be overcome by more complex statistical analysis like inverse probability weighting R. Two paragraphs were added (one in the Material and Method section and one in the Results section) to address the issue.
  2. Population and lesions. R. - Charlson's Comorbidity Index was provided. A brief sentence was added about conversion rate. - A picture was added to better describe the distribution of the lesions. Due to the multiplicity of the lesions, almost all the patients had at least one lesion deeply located or near a pedicle making difficult further comparison. - it is indeed possible that significative higher number of complications in postoperative course be related to more complex procedure afforded in open resection and it was stated. - The subdivision into CD grades was added and the specific complications were detailed in the text

  3. No details about the biology of disease. R. - In Table 5 there is also a line with "other molecular mutations" that includes BRAF, NRAS, MSS/MSI or other rarer mutations and it was stated also in the text. - Unfortunately, CEA values were not available for almost the half of the patients, precluding further evaluations. T and N status of the primary tumor were added in Table 2.

  4. Our burocracy is quite slow. Due to its retrospective nature the approval is quite sure, hopefully arriving soon.
  5. Tables were corrected and median with IQR values were reported. The citation was added. No R1 vascular were reported and it was stated.

Major changes are highlighted in yellow.

Reviewer 2 Report

In their study "A comparison between open and minimally invasive techniques for the resection of colorectal liver metastasis" the authors aimed to compare short, long-term outcomes of open and MIS in liver surgery as well as evaluate the safety of laparoscopic liver resections. They performed a prospective single centre observation analysis of clinical outcomes of all patients undergoing liver resections due to colorectal liver metastases between November 2018 and August 2021.

Major drawbacks:
1.     Small sample size - 10 cases in the robotic/laparoscopic group and 41 in the open group - major hepatobiliary centers in Germany report up to 300 cases per year including a rate of major resections between 40 and 60%.
2.     Not randomized - even with a low case number randomization should be possible. Furthermore, the kind of technique of surgery was chosen by the surgeons, where the majority of the minor liver resections were performed in the MIC group, in contrast only 37% of minor resections were performed in the open group.
3.     No differentiation between laparoscopic and robotic liver resections is made. As well as no differentiation between open minor and open major liver resections was done.
4.     Different types for parenchymal transection: CUSA for open and Bipo for Lap/Robot.
5.     Statistically different war the number of lesions between groups. Patients needing a major resection were more frequently in the open group and many more. 
6.     A considerably larger trial on a comparison between open vs laparoscopic as well as laparoscopic vs robotic was recently published.

In general, due to the above mentioned there is not possible to make a comparison from the trial between open and MIC techniques.

Minor drawbacks: 
1.     No Clavien-Dindo classification in Results.
2.     Mistake in Results (line 123): “Fifty patients were treated with an open technique..”
3.     
Not clear how many patients received perioperative chemotherapy or biological therapy.

4.     The conclusions are “general“, not based on the recent study.

Author Response

- All the major drawbacks are clearly stated in the limits.

However, the small number of the patients is also related to the necessity of analyzing a homogeneous group of patients (first occurrence of colorectal adenocarcinoma), the global number of liver resections in our Unit is obviously higher. Due to the small sample, further subgroup analyses seem unworthy. Some more recent references were added.

- Mistakes were corrected. A line with biological therapy was added in both the tables of pre and post-operative treatments.

Reviewer 3 Report

This manuscript compares open and minimally invasive surgery for liver metastases in colorectal cancer. There are some major limitations. The number of laparoscopically resected patients is very small, and consists of a selected group of surgically easier resections. Thus, this manuscript reflects the learning curve of the institution. This has to be clearly stated in the introduction and in the discussion.

Data on preoperative chemotherapy should be defined to preoperative therapy for the primary cancer and therapy befor liver resection.

Author Response

Thanks for your comments.

- The language was revised by an American native speaker (added in the Acknowledgment section)

- Thus, this manuscript reflects the learning curve of the institution. This has to be clearly stated in the introduction and in the discussion. R. A few sentences were added in the Introduction and in the Discussion sections to address the issue

- Data on preoperative chemotherapy should be defined to preoperative therapy for the primary cancer and therapy befor liver resection. R. Respectfully, we decide to report data about chemotherapy before liver surgery. Due to different modality presentations (Sync/met) and due to the different modality treatments (liver/bowel first, combined), the variables would have been too much and could be difficult to be well-explained.

Round 2

Reviewer 1 Report

I really appreciate the revised version of the paper. 
I found improvement in several parts, from the statistical analysis to result descriptions. The IPW analysis, despite at the basic level, gives inspiration for following papers that may be faced when a larger cohort will be available. The figure 2 added is remarkable, giving to the paper an increasing interest and comprehension.

Author Response

Thank you very much for your suggestions

that allowed us to improve the paper.

Thank you very much for your final appreciations.